# REWEIGHTED SOLUTIONS FOR WEIGHTED LOW RANK APPROXIMATION

## ABSTRACT

Weighted low rank approximation is an important yet computationally challenging primitive with applications ranging from statistical analysis, model compression, and signal processing. To cope with the NP-hardness of this problem, prior work considers heuristics, bicriteria, or parameterized tractable algorithms to solve this problem. In this work, we introduce a new relaxed solution to weighted low rank approximation which outputs a matrix that is not necessarily low rank, but can be stored using very few parameters and gives provable approximation guarantees when the weight matrix has low rank. Our central idea is to use the weight matrix itself to reweight a low rank solution, which gives an extremely simple algorithm with remarkable empirical performance in applications to model compression. Our algorithm also gives nearly optimal communication complexity bounds for a natural distributed problem associated with this problem, for which we show matching communication lower bounds. Together, our communication complexity bounds show that the rank of the weight matrix provably parameterizes the communication complexity of weighted low rank approximation. We also obtain the first relative error guarantees for feature selection with a weighted objective.

## 1 INTRODUCTION

The approximation of matrices by one of lower rank has been, and continues to be, one of the most intensely studied and applied computational problems in statistics, machine learning, signal processing, and beyond. The classical approach to this problem is to approximate matrices $\mathbf{A} \in \mathbb{R}^{n \times d}$ by a rank $k$ matrix $\tilde{\mathbf{A}} \in \mathbb{R}^{n \times d}$ that minimizes the Frobenius norm error

$$\|\mathbf{A} - \tilde{\mathbf{A}}\|_F^2 := \sum_{i=1}^n \sum_{j=1}^d |\mathbf{A}_{i,j} - \tilde{\mathbf{A}}_{i,j}|^2, \qquad \mathrm{rank}(\tilde{\mathbf{A}}) \le k.$$

This problem is solved by the singular value decomposition (SVD), which can be computed in polynomial time. We will write $\mathbf{A}_k$ to denote the optimal rank $k$ approximation to $\mathbf{A}$ in the Frobenius norm, and we will write $\mathbf{A}_{-k} := \mathbf{A} - \mathbf{A}_k$ to denote the error of this approximation.

While this simple choice often gives satisfactory results, this loss function treats all entries of the matrix uniformly when trying to fit $\tilde{\mathbf{A}}$, which may not exactly align with the practitioner's desires if some of the entries are more crucial to fit than others. If one additionally has such information available in the form of nonnegative weights $\mathbf{W}_{i,j} \ge 0$ that reflects some measure of *importance* of each of the entries $(i,j)$, then this can be encoded in the loss function as follows:

$$\|\mathbf{A} - \tilde{\mathbf{A}}\|_{\mathbf{W},F}^2 := \sum_{i=1}^n \sum_{j=1}^d \mathbf{W}_{i,j}^2 \cdot |\mathbf{A}_{i,j} - \tilde{\mathbf{A}}_{i,j}|^2, \qquad \mathrm{rank}(\tilde{\mathbf{A}}) \le k.$$

This problem is known as the *weighted low rank approximation (WLRA)* problem. We write $\mathbf{A} \circ \mathbf{B}$ to denote the entrywise product for two matrices $\mathbf{A}$ and $\mathbf{B}$, so we may also write

$$\|\mathbf{A} - \tilde{\mathbf{A}}\|_{\mathbf{W},F}^2 := \|\mathbf{W} \circ (\mathbf{A} - \tilde{\mathbf{A}})\|_F^2 = \|\mathbf{W} \circ \mathbf{A} - \mathbf{W} \circ \tilde{\mathbf{A}}\|_F^2$$

The incorporation of weights into the low rank approximation problem gives this computational problem an incredible versatility for use in a long history of applications starting with its use in

factor analysis in the early statistical literature Young (1941). A popular special case is the *matrix completion* problem Rennie & Srebro (2005); Candès & Tao (2010); Keshavan et al. (2010), where the weights $\mathbf{W} \in \{0,1\}^{n \times d}$ are binary and encode whether a given entry of $\mathbf{A}$ is observed or not. This primitive has been useful in the design of recommender systems Koren et al. (2009); Chen et al. (2015); Lee et al. (2016), and has been famously applied in the 2006 Netflix Prize problem. More generally, the weights $\mathbf{W}$ can be used to reflect the variance or number of samples obtained for each of the entries, so that more "uncertain" entries can influence the objective function less Anandan & Irani (2002); Srebro & Jaakkola (2003). In the past few years, weighted low rank approximation has also been used to improve model compression algorithms, especially those for large scale LLMs, based on low rank approximations of weight matrices by taking into account the importance of parameters Arora et al. (2016); Hsu et al. (2022); Hua et al. (2022). Given the rapid growth of large scale machine learning models, model compression techniques such as weighted low rank approximation are expected to bring high value to engineering efforts for these models. Other applications of weighted low rank approximation include ecology Robin et al. (2019); Kidzinski et al. (2022), background modeling Li et al. (2017); Dutta et al. (2018), computational biology Tuzhilina et al. (2022), and signal processing Shpak (1990); Lu et al. (1997).

Approximation algorithms have long been considered for efficient low rank approximation problems, and we formalize the approximation guarantee that we study in Definition 1.1.

**Definition 1.1** (Approximate weighted low rank approximation). *Let* $\mathbf{W} \in \mathbb{R}^{n \times d}$ *be nonnegative, let* $\mathbf{A} \in \mathbb{R}^{n \times d}$, *and let* $k \in \mathbb{N}$. *Then in the* $\kappa$-*approximate rank* $k$ *weighted low rank approximation problem, we seek to output a matrix* $\tilde{\mathbf{A}} \in \mathbb{R}^{n \times d}$ *such that*

$$\|\mathbf{A} - \tilde{\mathbf{A}}\|_{\mathbf{W},F} \leq \kappa \min_{\mathrm{rank}(\mathbf{A}') \leq k} \|\mathbf{A} - \mathbf{A}'\|_{\mathbf{W},F}.$$

In Definition 1.1, we have purposefully under-specified requirements on $\tilde{\mathbf{A}}$. Part of this is to cope with the computational difficulty of WLRA. Indeed, while we ideally would like $\tilde{\mathbf{A}}$ to have rank at most $k$, solving for even an approximate such solution (with $\kappa = (1 + 1/\mathrm{poly}(n))$) is an NP-hard problem Gillis & Glineur (2011). Furthermore, allowing for additional flexibility in the choice of $\tilde{\mathbf{A}}$ may still be useful as long as $\tilde{\mathbf{A}}$ satisfies some sort of "parameter reduction" guarantee. A common choice is to allow $\tilde{\mathbf{A}}$ to have rank $k' \geq k$ slightly larger than $k$, which is known as a *bicriteria* guarantee. In this work, we will show a new relaxation of the constraints on $\tilde{\mathbf{A}}$ that allows us to achieve new approximation guarantees for WLRA.

### 1.1 OUR RESULTS

We present our main contribution in Theorem 1.2, which gives a simple approach to WLRA, under the assumption that the weight matrix $\mathbf{W}$ has low rank. We note that this assumption is very natural and captures natural cases, for example when $\mathbf{W}$ has block structure, and has been motivated and studied in prior work Razenshteyn et al. (2016); Ban et al. (2019). We also empirically verify this assumption in our experiments. We defer a further discussion of the low rank $\mathbf{W}$ assumption to Section 1.1.3 as well as prior works Razenshteyn et al. (2016); Ban et al. (2019).

The algorithm (shown in Algorithm 1) that we propose is extremely simple: compute a rank $rk$ approximation of $\mathbf{W} \circ \mathbf{A}$, and then divide the result entrywise by $\mathbf{W}$. Note that if we exactly compute the low rank approximation step by an SVD, then the optimal rank $rk$ approximation $(\mathbf{W} \circ \mathbf{A})_{rk}$ given by the SVD requires only $(n + d)rk$ parameters to store, and $\mathbf{W}$ also only requires $nr$ parameters to store. Thus, the solution $\mathbf{W}^{\circ -1} \circ (\mathbf{W} \circ \mathbf{A})_{rk}$ can be stored in a total of $O((n + d)rk)$ parameters, which is nearly optimal for constant rank $r = O(1)$.

---

**Algorithm 1** Weighted low rank approximation

**input:** input matrix $\mathbf{A} \in \mathbb{R}^{n \times d}$, nonnegative weights $\mathbf{W} \in \mathbb{R}^{n \times d}$ with rank $r$.
**output:** approximate solution $\tilde{\mathbf{A}}$.

1: Compute a rank $rk$ approximation $\tilde{\mathbf{A}}_{\mathbf{W}}$ of $\mathbf{W} \circ \mathbf{A}$
2: **return** $\tilde{\mathbf{A}} := \mathbf{W}^{\circ -1} \circ \tilde{\mathbf{A}}_{\mathbf{W}}$

---

While our discussion thus far has simply used the SVD to compute the rank $rk$ approximation $(\mathbf{W} \circ \mathbf{A})_{rk}$, we obtain other useful guarantees by allowing for approximate solutions $\tilde{\mathbf{A}}_{\mathbf{W}}$ that only approximately minimize $\|\mathbf{W} \circ \mathbf{A} - \tilde{\mathbf{A}}_{\mathbf{W}}\|_F$. For example, by computing the rank $rk$ approximation $\tilde{\mathbf{A}}_{\mathbf{W}}$ using faster randomized approximation algorithms for the SVD Clarkson & Woodruff (2013); Musco & Musco (2015); Avron et al. (2017), we obtain algorithms for WLRA with similar running time. In general, we prove the following theorem:

**Theorem 1.2.** *Let $\mathbf{W} \in \mathbb{R}^{n \times d}$ be a nonnegative weight matrix with rank $r$. Let $\mathbf{A} \in \mathbb{R}^{n \times d}$ and let $k \in \mathbb{N}$. Suppose that $\tilde{\mathbf{A}}_{\mathbf{W}} \in \mathbb{R}^{n \times d}$ satisfies*

$$\|\mathbf{W} \circ \mathbf{A} - \tilde{\mathbf{A}}_{\mathbf{W}}\|_F^2 \leq \kappa \min_{\mathrm{rank}(\mathbf{A}') \leq rk} \|\mathbf{W} \circ \mathbf{A} - \mathbf{A}'\|_F^2 = \kappa \|(\mathbf{W} \circ \mathbf{A})_{-rk}\|_F^2$$

*and let $\tilde{\mathbf{A}} := \mathbf{W}^{\circ -1} \circ \tilde{\mathbf{A}}_{\mathbf{W}}$, where $\mathbf{W}^{\circ -1} \in \mathbb{R}^{n \times d}$ denotes the entrywise inverse of $\mathbf{W}$. Then,*

$$\|\mathbf{A} - \tilde{\mathbf{A}}\|_{\mathbf{W}, F}^2 \leq \kappa \min_{\mathrm{rank}(\mathbf{A}') \leq k} \|\mathbf{A} - \mathbf{A}'\|_{\mathbf{W}, F}^2$$

*In particular, we obtain a solution with $\kappa = (1 + \varepsilon)$ in running time $O(\mathrm{nnz}(\mathbf{A})) + \tilde{O}(n(rk)^2/\varepsilon + \mathrm{poly}(rk/\varepsilon))$ by using randomized low rank approximation algorithms of Avron et al. (2017).*

We prove Theorem 1.2 in Section 2.

We note that as stated, the approximation given by Algorithm 1 may not always be desirable, since in general, $\mathbf{W}^{\circ -1}$ cannot be computed without multiplying out the low rank factors of $\mathbf{W}$. However, we show in Lemma A.1 that for a broad family of structured matrices formed by the sum of support-disjoint rank 1 matrices and a sparse matrix, $\mathbf{W}^{\circ -1}$ can in fact be stored and applied in the same time as $\mathbf{W}$. These capture a large number of commonly used weight matrix patterns in practice, such as Low-Rank Plus Sparse, Low-Rank Plus Diagonal, Low-Rank Plus Block Diagonal, Monotone Missing Data Pattern, and Low-Rank Plus Banded matrices Musco et al. (2021).

For general matrices, we present a more practical alternative in Algorithm 2 where we compute a low rank approximation of $\mathbf{W}^{\circ -1}$, so that the entrywise inverse can be applied efficiently.

---

**Algorithm 2** Weighted low rank approximation (practical)

---

**input:** input matrix $\mathbf{A} \in \mathbb{R}^{n \times d}$, nonnegative weights $\mathbf{W} \in \mathbb{R}^{n \times d}$ with rank $r$.
**output:** approximate solution $\tilde{\mathbf{A}}$.

1: Compute a rank $rk$ approximation $\tilde{\mathbf{W}}_{\mathrm{inv}}$ of $\mathbf{W}^{\circ -1}$
2: Compute a rank $rk$ approximation $\tilde{\mathbf{A}}_{\mathbf{W}}$ of $\tilde{\mathbf{W}}_{\mathrm{inv}}^{\circ -1} \circ \mathbf{A}$
3: **return** $\tilde{\mathbf{A}} := \tilde{\mathbf{W}}_{\mathrm{inv}} \circ \tilde{\mathbf{A}}_{\mathbf{W}}$

---

### 1.1.1 COLUMN SUBSET SELECTION FOR WEIGHTED LOW RANK APPROXIMATION

Another advantage of allowing for approximation algorithms for computing low rank approximations to $\mathbf{W} \circ \mathbf{A}$ is that we can employ *column subset selection* approaches to low rank approximation Frieze et al. (2004); Deshpande & Vempala (2006); Drineas et al. (2006; 2008); Boutsidis et al. (2016); Altschuler et al. (2016). That is, it is known that the Frobenius norm low rank approximation problem admits $(1 + \varepsilon)$-approximate low rank approximations whose left factor is formed by a subset of at most $O(k/\varepsilon)$ columns of the input matrix. In particular, these results show the existence of approximate solutions to the low rank approximation problem that preserves the sparsity of the input matrix, and thus can lead to a reduced solution size when the input matrix has sparse columns. Furthermore, column subset selection solutions to low rank approximation give a natural approach for *unsupervised feature selection*. Thus, as a corollary of Theorem 1.2, we obtain the first relative error guarantee for unsupervised feature selection with a weighted Frobenius norm objective. Weaker additive error guarantees were previously studied by Dai (2023); Axiotis & Yasuda (2023)[1].

**Corollary 1.3** (Column subset selection for weighted low rank approximation)**.** *There is an algorithm that computes a subset $S \subseteq [d]$ of $|S| = O(rk/\varepsilon)$ columns and $\mathbf{X} \in \mathbb{R}^{|S| \times d}$ such that*

$$\left\|\mathbf{A} - \mathbf{W}^{\circ -1} \circ ((\mathbf{W} \circ \mathbf{A})|^S \mathbf{X})\right\|_{\mathbf{W}, F}^2 \leq (1 + \varepsilon) \min_{\mathrm{rank}(\mathbf{A}') \leq k} \|\mathbf{A} - \mathbf{A}'\|_{\mathbf{W}, F}^2$$

---

[1]The result of Dai (2023) contained an error, which we correct, tighten, and simplify in Appendix D.

*where for a matrix* $\mathbf{B} \in \mathbb{R}^{n \times d}$, $\mathbf{B}|^S$ *denotes the matrix formed by the columns of* $\mathbf{B}$ *indexed by* $S$.

*Proof.* This follows from Theorem 1.2 by computing the rank $rk$ approximation $\tilde{\mathbf{A}}_{\mathbf{W}}$ to $\mathbf{W} \circ \mathbf{A}$ via column subset selection algorithms given by, e.g., Boutsidis et al. (2016). □

Note that in Corollary 1.3, the approximation $\mathbf{W}^{\circ -1} \circ ((\mathbf{W} \circ \mathbf{A})|^S \mathbf{X})$ only depends on $\mathbf{A}$ through the columns $\mathbf{A}|^S$, and thus giving an approach to column subset selection with a weighted objective.

### 1.1.2 NEARLY OPTIMAL COMMUNICATION COMPLEXITY BOUNDS

As a consequence of Corollary 1.3, we obtain another important result for WLRA in the setting of *communication complexity*. Here, we obtain nearly optimal communication complexity bounds for constant factor approximations (i.e. $\kappa = O(1)$) to distributed WLRA for a wide range of parameters. While many works have studied distributed LRA in depth (Sarlós, 2006; Clarkson & Woodruff, 2009; 2013; Macua et al., 2010; Kannan et al., 2014; Ghashami et al., 2016; Boutsidis et al., 2016), we are surprisingly the first to initiate a study of this problem for WLRA.

The communication setting we consider is as follows. We have two players, Alice and Bob, where Alice has an input matrix $\mathbf{A}$ and would like to communicate an approximate WLRA solution to Bob. Communication complexity is of great interest in modern computing, where exchanging bits can be a critical bottleneck in large scale computation. While we consider two players in this discussion for simplicity, our algorithms also apply to a distributed computing setting, where the columns of the input matrix are partitioned among $m$ servers as $m$ matrices $\mathbf{A}^{(1)}, \mathbf{A}^{(2)}, \ldots, \mathbf{A}^{(m)}$, and some central coordinator outputs a WLRA of the concatenation $\mathbf{A} = [\mathbf{A}^{(1)}, \mathbf{A}^{(2)}, \ldots, \mathbf{A}^{(m)}]$ of these columns.

**Definition 1.4** (WLRA: communication game). *Let Alice and Bob be two players. Let* $\mathbf{W} \in \mathbb{Z}^{n \times d}$ *be nonnegative, let* $\mathbf{A} \in \mathbb{Z}^{n \times d}$, *and let* $k \in \mathbb{N}$. *Furthermore, let* $\mathbf{W}$ *and* $\mathbf{A}$ *have entries at most* $\mathbf{W}_{i,j}, |\mathbf{A}_{i,j}| \leq \mathrm{poly}(nd)$. *We let both Alice and Bob receive the weight matrix* $\mathbf{W}$ *as input, and we give only Alice the input matrix* $\mathbf{A}$. *We say that Alice and Bob solve the* $\kappa$-approximate rank $k$ weighted low rank approximation communication game *using* $B$ *bits of communication if Alice sends at most* $B$ *bits to Bob, and Bob outputs any* matrix $\tilde{\mathbf{A}} \in \mathbb{R}^{n \times d}$ *satisfying*

$$\|\mathbf{A} - \tilde{\mathbf{A}}\|_{\mathbf{W},F} \leq \kappa \min_{\mathrm{rank}(\mathbf{A}') \leq k} \|\mathbf{A} - \mathbf{A}'\|_{\mathbf{W},F}.$$

Suppose that $\mathbf{A}$ has columns which each have at most $s$ nonzero entries. Then, the solution given by Corollary 1.3 can be communicated to Bob using just $O(srk/\varepsilon + rkd)$ numbers ($O(srk/\varepsilon)$ for the $O(rk/\varepsilon)$ columns of $\mathbf{A}$ and $O(rkd)$ for $\mathbf{X}$), or $O((srk/\varepsilon + rkd)\log(nd))$ bits under our bit complexity assumptions. Thus, when the number of columns $d$ is at most the column sparsity $s$, then we obtain an algorithm which uses only $O((srk/\varepsilon)\log(nd))$ bits of communication. More generally, if the columns of $\mathbf{A}$ are distributed among $m$ servers, then a solution can be computed using $O((msrk/\varepsilon)\log(nd))$ bits of communication by using work of Boutsidis et al. (2016).

In fact, we show a nearly matching communication lower bound. In particular, we show that $\Omega(srk)$ bits of communication is required to output *any* matrix (not necessarily structured) that achieves a weighted Frobenius norm loss that is *any* finite factor within the optimal solution. Our lower bound is information theoretic, and also immediately implies an $\Omega(msrk)$ bit lower bound in the distributed setting of $m$ servers if each server must output a solution, as considered by Boutsidis et al. (2016).

**Theorem 1.5.** *Let* $\mathbf{W}$ *be a binary block diagonal mask (Definition 3.2) and let* $k \in \mathbb{N}$. *Suppose that a randomized algorithm solves, for every* $\mathbf{C} \in \mathbb{Z}^{n \times n}$ *with at most* $s$ *nonzero entries in each column, the* $\kappa$-approximate weighted low rank approximation problem using $B$ bits of communication with probability at least* $2/3$, *for any* $1 \leq \kappa < \infty$. *If* $s, k \leq n/r$, *then* $B = \Omega(srk)$.

By proving a nearly tight communication complexity bound of $\tilde{\Theta}(rsk)$ for computing constant factor WLRAs, we arrive at the following qualitative observation: *the rank $r$ of the weight matrix* $\mathbf{W}$ *parameterizes the communication complexity of WLRA*. Note that a similar conclusion was drawn for the *computational* complexity of WLRA in the work of Razenshteyn et al. (2016), where it was shown that WLRA is fixed parameter tractable in the parameter $r$, and also must have running time exponential in $r$ under natural complexity theoretic assumptions. Thus, we believe that one important contribution of our work is to provide further evidence, both empirical and theoretical, that the rank $r$ of the weight matrix $\mathbf{W}$ is a natural parameter to consider when studying WLRA.

### 1.1.3 EXPERIMENTS

We demonstrate the empirical performance of our WLRA algorithms through experiments for model compression tasks. This application of WLRA was suggested by Hsu et al. (2022); Hua et al. (2022), which we find to be a particularly relevant application of weighted low rank approximation due to the trend of extremely large models. In the model compression setting, we wish to approximate the hidden layer weight matrices of neural networks by much smaller matrices. A classical way to do this is to use low rank approximation (Sainath et al., 2013; Kim et al., 2016; Chen et al., 2018). While this often gives reasonable results, the works of Hsu et al. (2022); Hua et al. (2022) show that significant improvements can be obtained by taking into account the importance of each of the parameters in the LRA problem. We thus conduct our experiments in this setting.

We first show in Section 4.1 that the importance matrices arising this application are indeed very low rank. We may interpret this phenomenon intuitively: we hypothesize that the importance score of some parameter $\mathbf{A}_{i,j}$ is essentially the product of the importance of the corresponding input $i$ and the importance of the corresponding output $j$. This observation may be of independent interest, and also empirically justifies the low rank weight matrix assumption that we make in this work, as well as works of Razenshteyn et al. (2016); Ban et al. (2019). While WLRA with a rank 1 weight matrix is known to be solvable efficiently via the SVD, our result shows that general low rank weight matrices also yield efficient algorithms via the SVD.

Next in Section 4.2, we conduct experiments which demonstrates the superiority of our methods in practice. Of the algorithms that we compare to, an expectation-minimization (EM) approach of Srebro & Jaakkola (2003) gives the smallest loss albeit with a very high running time, and our algorithm nearly matches this loss with an order of magnitude lower running time. We also show that this solution can be refined with EM, producing the best trade-off between efficiency and accuracy. One of the baselines we compare is a sampling algorithm of Dai (2023), whose analysis contains an error which we correct, simplify, and tighten.

## 1.2 RELATED WORK

We survey a number of related works on approximation algorithms for weighted low rank approximation. One of the earliest algorithms for this problem is a natural EM approach proposed by Srebro & Jaakkola (2003). Another related approach is to parameterize the low rank approximation $\tilde{\mathbf{A}}$ as the product $\mathbf{UV}$ of two matrices $\mathbf{U} \in \mathbb{R}^{n \times k}$ and $\mathbf{V} \in \mathbb{R}^{k \times d}$ and alternately minimize the two matrices, known as *alternating least squares*. This algorithm has been studied in a number of works (Hastie et al., 2015; Li et al., 2016; Song et al., 2023). The work of Bhaskara et al. (2021) proposes an approach to weighted low rank approximation based on a greedy pursuit, where rank one factors are iteratively added based on an SVD of the gradient matrix. Finally, fixed parameter tractable algorithms have been considered in Razenshteyn et al. (2016); Ban et al. (2019) based on sketching techniques.

## 2 APPROXIMATION ALGORITHMS

The following simple observation is the key idea behind Theorem 1.2:

**Lemma 2.1.** *Let* $\mathbf{W}, \mathbf{A}' \in \mathbb{R}^{n \times d}$ *with* $\mathrm{rank}(\mathbf{W}) \leq r$ *and* $\mathrm{rank}(\mathbf{A}') \leq k$. *Then,* $\mathrm{rank}(\mathbf{W} \circ \mathbf{A}') \leq rk$.

*Proof.* Since $\mathrm{rank}(\mathbf{W}) \leq k$, it can be written as $\mathbf{W} = \sum_{i=1}^{r} \mathbf{u}_i \mathbf{v}_i^\top$ for $\mathbf{u}_i \in \mathbb{R}^n$ and $\mathbf{v}_i \in \mathbb{R}^d$. Then,

$$\mathbf{W} \circ \mathbf{A}' = \sum_{i=1}^{r} (\mathbf{u}_i \mathbf{v}_i^\top) \circ \mathbf{A}' = \sum_{i=1}^{r} \mathrm{diag}(\mathbf{u}_i) \mathbf{A}' \, \mathrm{diag}(\mathbf{v}_i)$$

so $\mathbf{W} \circ \mathbf{A}'$ is the sum of $r$ matrices, each of which is rank $k$. Thus, $\mathbf{W} \circ \mathbf{A}'$ has rank at most $rk$. $\square$

Using Lemma 2.1, we obtain the following:

**Theorem 1.2.** *Let* $\mathbf{W} \in \mathbb{R}^{n \times d}$ *be a nonnnegative weight matrix with rank* $r$. *Let* $\mathbf{A} \in \mathbb{R}^{n \times d}$ *and let* $k \in \mathbb{N}$. *Suppose that* $\tilde{\mathbf{A}}_{\mathbf{W}} \in \mathbb{R}^{n \times d}$ *satisfies*

$$\|\mathbf{W} \circ \mathbf{A} - \tilde{\mathbf{A}}_{\mathbf{W}}\|_F^2 \leq \kappa \min_{\mathrm{rank}(\mathbf{A}') \leq rk} \|\mathbf{W} \circ \mathbf{A} - \mathbf{A}'\|_F^2 = \kappa \|(\mathbf{W} \circ \mathbf{A})_{-rk}\|_F^2$$

*and let* $\tilde{\mathbf{A}} := \mathbf{W}^{\circ-1} \circ \tilde{\mathbf{A}}_{\mathbf{W}}$, *where* $\mathbf{W}^{\circ-1} \in \mathbb{R}^{n \times d}$ *denotes the entrywise inverse of* $\mathbf{W}$. *Then,*

$$\|\mathbf{A} - \tilde{\mathbf{A}}\|_{\mathbf{W},F}^2 \leq \kappa \min_{\text{rank}(\mathbf{A}') \leq k} \|\mathbf{A} - \mathbf{A}'\|_{\mathbf{W},F}^2$$

*In particular, we obtain a solution with* $\kappa = (1 + \varepsilon)$ *in running time* $O(\text{nnz}(\mathbf{A})) + \tilde{O}(n(rk)^2/\varepsilon + \text{poly}(rk/\varepsilon))$ *by using randomized low rank approximation algorithms of* Avron et al. (2017).

*Proof.* Note that $\|\mathbf{W}^{\circ-1} \circ \tilde{\mathbf{A}}_{\mathbf{W}} - \mathbf{A}\|_{\mathbf{W},F}^2 = \|\tilde{\mathbf{A}}_{\mathbf{W}} - \mathbf{W} \circ \mathbf{A}\|_F^2$, which is at most $\kappa \|(\mathbf{W} \circ \mathbf{A})_{-rk}\|_F^2$ by assumption. On the other hand for any rank $k$ matrix $\mathbf{A}'$, $\|\mathbf{A}' - \mathbf{A}\|_{\mathbf{W},F} = \|\mathbf{W} \circ \mathbf{A}' - \mathbf{W} \circ \mathbf{A}\|_F$ can be lower bounded by $\|(\mathbf{W} \circ \mathbf{A})_{-rk}\|_F$ since $\mathbf{W} \circ \mathbf{A}'$ has rank at most $rk$ by Lemma 2.1. Thus,

$$\|\mathbf{W}^{\circ-1} \circ \tilde{\mathbf{A}}_{\mathbf{W}} - \mathbf{A}\|_{\mathbf{W},F}^2 \leq \kappa \|(\mathbf{W} \circ \mathbf{A})_{-rk}\|_F^2 \leq \kappa \min_{\text{rank}(\mathbf{A}') \leq k} \|\mathbf{A} - \mathbf{A}'\|_{\mathbf{W},F}^2. \qquad \square$$

## 3 COMMUNICATION COMPLEXITY BOUNDS

We show that our approach to weighted low rank approximation in Theorem 1.2 gives nearly optimal bounds for this problem in the setting of communication complexity.

Our first result is an upper bound for the communication game in Definition 1.4.

**Theorem 3.1.** *Let* $\mathbf{W} \in \mathbb{Z}^{n \times d}$ *be a nonnegative rank* $k$ *weight matrix and let* $\mathbf{A} \in \mathbb{Z}^{n \times d}$ *be an input matrix with at most* $s$ *nonzero entries in each column. There is an algorithm which solves the* $(1 + \varepsilon)$-*approximate weighted low rank approximation communication game (Definition 1.4) using at most* $B = O((srk/\varepsilon + rkd) \log(nd))$ *bits of communication.*

*Proof.* The algorithm is simply to use the column subset selection-based WLRA algorithm of Corollary 1.3 and then to send the columns of $\mathbf{A}$ indexed by the column subset $S$ and $\mathbf{X}$. $\qquad \square$

On the other hand, we show a communication complexity lower bound showing that the number of bits $B$ exchanged by Alice and Bob must be at least $\Omega(rsk)$. Our lower bound holds even when the weight matrix $\mathbf{W}$ is the following simple binary matrix.

**Definition 3.2** (Block diagonal mask). *Let* $r \in \mathbb{N}$ *and let* $n$ *be an integer multiple of* $r$. *Then,* $\mathbf{W} \in \{0, 1\}^{n \times n}$ *is the* block diagonal mask *associated with these parameters if* $\mathbf{W}$ *is the* $r \times r$ *block diagonal matrix with diagonal blocks given by the* $n/r \times n/r$ *all ones matrix and off-diagonal blocks given by the* $n/r \times n/r$ *all zeros matrix.*

We give our communication complexity lower bound in the following theorem.

**Theorem 1.5.** *Let* $\mathbf{W}$ *be a binary block diagonal mask (Definition 3.2) and let* $k \in \mathbb{N}$. *Suppose that a randomized algorithm solves, for every* $\mathbf{C} \in \mathbb{Z}^{n \times n}$ *with at most* $s$ *nonzero entries in each column, the* $\kappa$-*approximate weighted low rank approximation problem using* $B$ *bits of communication with probability at least* $2/3$, *for any* $1 \leq \kappa < \infty$. *If* $s, k \leq n/r$, *then* $B = \Omega(srk)$.

*Proof.* Let $\mathbf{A}_{\text{dense}} \in \{0, 1\}^{sr \times k}$ be a uniformly random binary matrix, and let $\mathbf{A}_{\text{pad}} \in \{0, 1\}^{n \times n/r}$ be formed by padding the columns of $\mathbf{A}_{\text{dense}}$ with $n/r - k$ columns of all zeros and padding each block of $s$ contiguous rows with $n/r - s$ rows of all zeros. For $j \in [r]$, let $\mathbf{A}_{\text{pad}}^{(j)}$ denote the restriction of $\mathbf{A}_{\text{pad}}$ to the $j$th contiguous block of $n/r$ rows. We then construct $\mathbf{A} \in \mathbb{R}^{n \times n}$ by horizontally concatenating $r$ copies of $\mathbf{A}_{\text{pad}}$.

Note that an optimal rank $k$ approximation can achieve $0$ loss in the $\mathbf{W}$-weighted Frobenius norm. Indeed, we can take $\mathbf{A}^*$ to be the horizontal concatenation of $r$ copies of $\mathbf{A}_{\text{pad}}$. Since $\mathbf{A}_{\text{pad}}$ has rank $k$, $\mathbf{A}^*$ also has rank $k$. Furthermore, on the $j$-th nonzero blocks of $\mathbf{W}$, $\mathbf{A}_{\text{pad}}$ has the same entries as $\mathbf{A}_{\text{pad}}^{(j)}$. Thus, it follows that an approximation $\tilde{\mathbf{A}}$ that achieves *any* finite approximation factor $\kappa$ must exactly recover $\mathbf{A}$, restricted to the support of $\mathbf{W}$. In turn, this means that such an approximation $\tilde{\mathbf{A}}$ can also be used to recover $\mathbf{A}_{\text{dense}}$.

It now follows by a standard information theoretic argument that $B = \Omega(srk)$ (see Appendix B for further details). $\qquad \square$

## 4 EXPERIMENTS

As discussed in Section 1.1.3, we conduct experiments for weighted low rank approximation in the setting of model compression (as proposed by Hsu et al. (2022); Hua et al. (2022)). In our experiments, we train a basic multilayer perceptron (MLP) on four image datasets, `mnist`, `fashion_mnist`, `smallnorb`, and `colorectal_histology` which were selected from the `tensorflow_datasets` catalogue for simplicity of processing (e.g. fixed feature size, no need for embeddings, etc). We then compute a matrix of importances of each of the parameters in a hidden layer of the MLP given by the Fisher information matrix. Finally, we compute a weighted low rank approximation of the hidden layer matrix using the Fisher information matrix as the weights $\mathbf{W}$.

Our experiments are conducted on a 2019 MacBook Pro with a 2.6 GHz 6-Core Intel Core i7 processor. All code used in the experiments are available in the supplement.

Table 1: Datasets used in experiments

| Dataset | Image dim. | Flattened dim. | Neurons | Matrix dim. |
|---|---|---|---|---|
| mnist | $(28, 28, 1)$ | 784 | 128 | $784 \times 128$ |
| fashion_mnist | $(28, 28, 1)$ | 784 | 128 | $784 \times 128$ |
| smallnorb | $(96, 96, 1)$ | 9216 | 1024 | $9216 \times 1024$ |
| colorectal_histology | $(150, 150, 3)$ | 67500 | 1024 | $67500 \times 1024$ |

### 4.1 THE LOW RANK WEIGHT MATRIX ASSUMPTION IN PRACTICE

We first demonstrate that for the task of model compression, the weight matrix is approximately low rank in practice. The weight matrix $\mathbf{W}$ in this setting is the empirical Fisher information matrix of the hidden layer weights $\mathbf{A}$, where the empirical Fisher information of the $(i, j)$-th entry $\mathbf{A}_{i,j}$ is given by

$$\mathbf{W}_{i,j} := \mathbb{E}_{\mathbf{x} \sim \mathcal{D}} \left[ \left( \frac{\partial}{\partial \mathbf{A}_{i,j}} \mathcal{L}(\mathbf{x}; \mathbf{A}) \right)^2 \right]$$

where $\mathcal{L}(\mathbf{x}; \mathbf{A})$ denotes the loss of the neural network on the data point $\mathbf{x}$ and hidden layer weights $\mathbf{A}$, and $\mathcal{D}$ denotes the empirical distribution (that is, the uniform distribution over the training data).

Plots of the empirical Fisher matrix (Figure 1) reveal nontrivial low rank structure to the matrices, and the spectrum of the Fisher matrix confirms that the vast majority of the Frobenius norm is contained in the first singular value (Table 2). We also plot the spectrum itself in Figure 5 in the appendix.

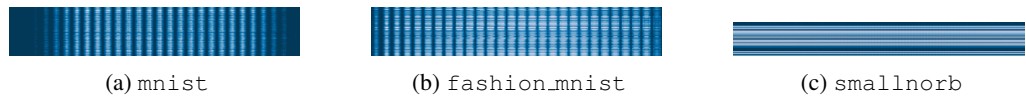

(a) `mnist`  (b) `fashion_mnist`  (c) `smallnorb`

Figure 1: Low rank structure of Fisher weight matrices

Table 2: Singular value distribution of Fisher matrix

| Dataset | % Frobenius mass in first singular value |
|---|---|
| mnist | 95.4% |
| fashion_mnist | 95.9% |
| smallnorb | 99.9% |
| colorectal_histology | 99.3% |

## 4.2 Approximation quality and running time

In this section, we compare the performance of our Algorithm 2 (denoted as `svd_w` in the following discussion) with a variety of previously proposed algorithms for weighted low rank approximation.

We consider the following algorithms: `adam`, `em`, `greedy`, `sample`, and `svd`, which we next explain in detail. We first consider `adam`, in which we simply parameterize the WLRA problem as an optimization problem in the factorized representation $\mathbf{UV}$ for factors $\mathbf{U} \in \mathbb{R}^{n \times k}$ and $\mathbf{V} \in \mathbb{R}^{k \times d}$, and optimize this loss function using the Adam optimizer provided in the `tensorflow` library. Such an approach is well-studied for the standard low rank approximation problem Li et al. (2018); Ye & Du (2021), and empirically performs well for weighted low rank approximation as well. This was run for 100 epochs, with an initial learning rate of 1.0 decayed by a factor of 0.7 every 10 steps. The `em` algorithm was proposed by Srebro & Jaakkola (2003) for the WLRA problem, and involves iteratively "filling in" missing values and recomputing a low rank approximation. In the experiments, we run 25 iterations. The `greedy` algorithm is a greedy basis pursuit algorithm proposed by Bhaskara et al. (2021) and iteratively adds new directions to the low rank approximation by taking an SVD of the gradient of the objective. Similar algorithms were also studied in Shalev-Shwartz et al. (2011); Khanna et al. (2017); Axiotis & Sviridenko (2021) for general rank-constrained convex optimization problems. The `sample` algorithm is a row norm sampling approach studied by Dai (2023). Finally, `svd` simply computes an SVD of the original matrix $\mathbf{W}$, without regard to the weights $\mathbf{W}$.

We compute low rank approximations for ranks 1 through 20 on four datasets, and plot the loss and the running time against the rank in Figures 2 and 3, respectively. The values in the figures are tabluated at ranks 20, 10, and 5 in Tables 3, 4, and 5 in the supplement. We observe that our `svd_w` algorithm performs among the best in the approximation loss (Figure 2), nearly matching the approximation quality achieved by much more computational expensive algorithms such as `adam` and `em`, while requiring much less computational time (Figure 3).

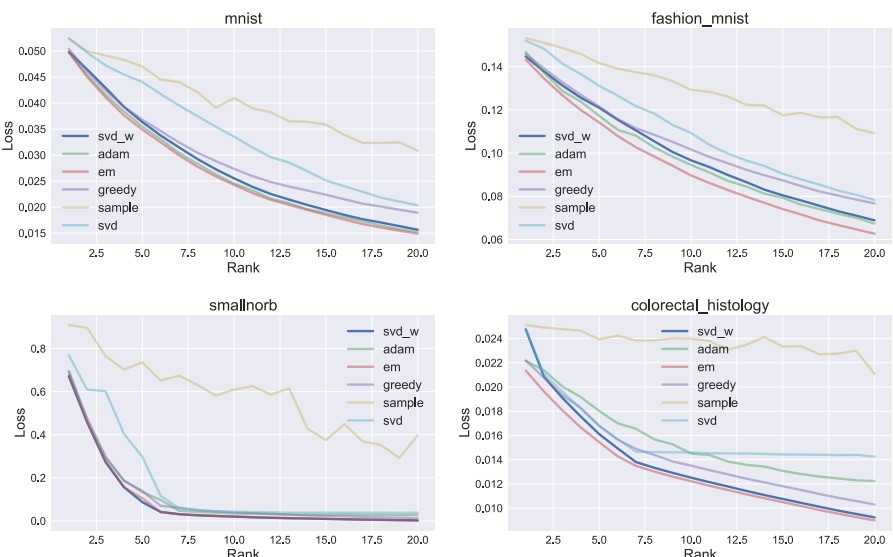

Figure 2: Fisher-weighted low rank approximation loss of weighted low rank approximation algorithms for model compression four datasets. Results are averaged over 5 trials.

While in some cases the `em` algorithm may eventually produce a better solution, we note that our `svd_w` may be improved by initializing the `em` algorithm with this solution, which produces an algorithm which quickly produces a superior solution with much fewer iterations (Figure 4).

## 5 Conclusion

In this work, we studied new algorithms for the weighted low rank approximation problem, which has countless applications in statistics, machine learning, and signal processing. We propose an approach

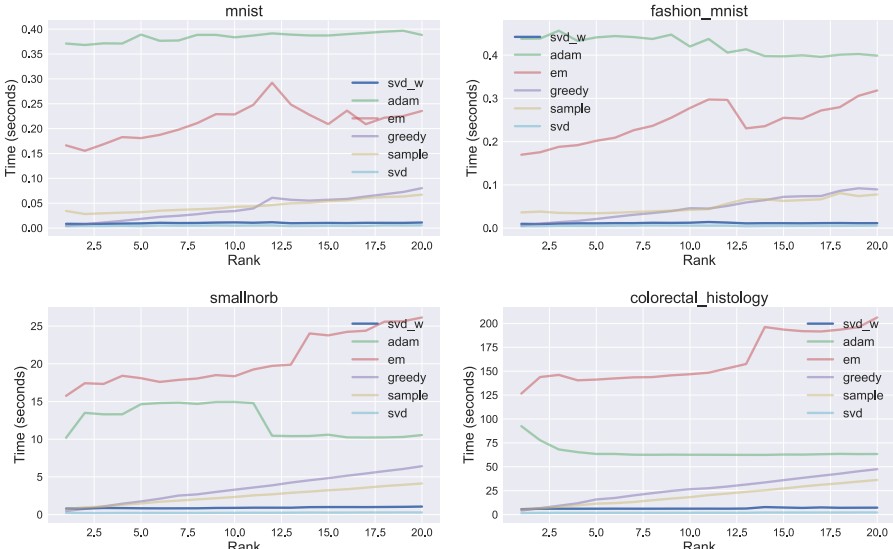

Figure 3: Running time of weighted low rank approximation algorithms for model compression four datasets. Results are averaged over 5 trials.

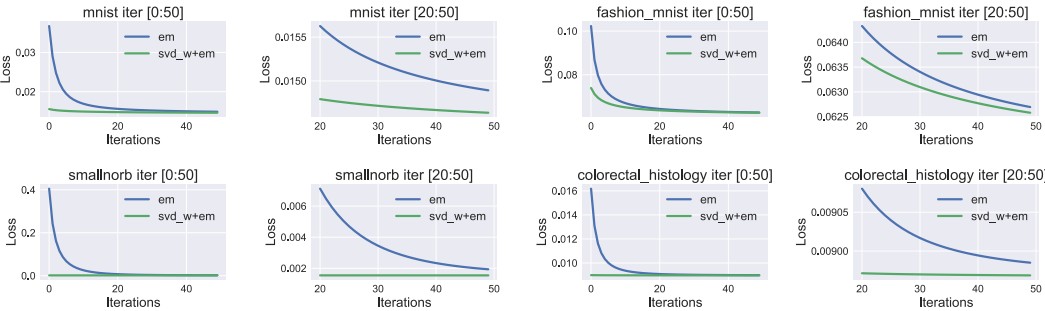

Figure 4: Improving the `svd_w` solution with `em` iterations for a rank 20 approximation.

based on reweighting a low rank matrix, which is a novel class of relaxed solutions to the WLRA problem, and give provable guarantees under the assumption that the weight matrix $\mathbf{W}$ has low rank. Theoretically, this allows us to obtain an algorithm for WLRA with nearly optimal communication complexity, for which we show nearly matching communication complexity lower bounds, which shows that the rank of the weight matrix tightly parameterizes the communication complexity of this problem. We also give the first guarantees for column subset selection for weighted low rank approximation, which gives a notion of feature selection with a weighted objective. Finally, we show that in practice, our approach gives a highly efficient algorithm that outperforms prior algorithms for WLRA, particularly when combined with refinement using expectation-maximization.

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
