# OpenReview forum: "Reweighted Solutions for Weighted Low Rank Approximation"
_ICLR.cc/2024/Conference — Submitted to ICLR 2024_

### Official Review · Reviewer_ftz8 · 2023-10-24

**Soundness:** 2 fair
**Presentation:** 2 fair
**Contribution:** 2 fair
**Rating:** 5
**Confidence:** 3

**Summary:**

The paper studies the weighted low rank approximation problem (WLRA) when the weight matrix is also of a low rank. The authors propose a new relaxed solution to this problem which outputs a matrix that can be well stored. As a corollary, the authors also give nearly optimal communication complexity bounds for another distributed problem.

**Strengths:**

- The studied problem of WLRA is important.
- The proposed algorithm is efficient and can be implemented.
- The writing is clear.

**Weaknesses:**

- The output of the main algorithm is not guaranteed to be of low rank, which is inconsistent with the original goal of WLRA. It may be better to discuss more about this type of output, specifically, why it can replace a low-rank matrix in practice.
- There is no theoretical guarantee for Algorithm 2, which makes it less interesting.

**Questions:**

- On page 2 before Algorithm 1, the authors claim that the storage space $O((n+d)rk)$ is nearly optimal for $r=O(1)$. Does there a matching lower bound of space for WLRA when $r=O(1)$?
- Is rank $r$ known in advance? If not, suppose we run Algorithm 1 with $r > rank(W)$, what happens?

---

> ### Author Response · Authors · 2023-11-18
> **Thank you for your review!**
>
> > The output of the main algorithm is not guaranteed to be of low rank, which is inconsistent with the original goal of WLRA. It may be better to discuss more about this type of output, specifically, why it can replace a low-rank matrix in practice.
>
> We have indeed departed from outputting a low rank matrix for the WLRA problem, and instead opted for a generalized solution. However, this solution can still be stored using few parameters, and can also be applied quickly if we use our practical variation of Algorithm 2. Our extensive experiments show that these relaxed solutions are indeed useful in practice, and demonstrates that it allows for solutions that can be stored and applied efficiently, allowing for state of the art performance for tasks such as neural network compression with a weighted objective.
>
> > There is no theoretical guarantee for Algorithm 2, which makes it less interesting.
>
> While we have not been able to analyze Algorithm 2, our experiments demonstrate its effectiveness in practice. We believe this is already compelling evidence for practitioners to adopt our approach.
>
> > On page 2 before Algorithm 1, the authors claim that the storage space $O((n+d)rk)$ is nearly optimal for $r = O(1)$. Does there a matching lower bound of space for WLRA when $r = O(1)$?
>
> When $r = O(1)$, the storage space is $O((n+d)k)$. Note that the size of the output requires this many bits of space, since the output is a factorization of the $n\times d$ matrix into a $n\times k$ matrix and a $k\times d$ matrix.
>
> > Is rank $r$ known in advance? If not, suppose we run Algorithm 1 with $r > \mathrm{rank}(\mathbf W)$, what happens?
>
> We assume that $r$ is known in advance. Note that $r$ can be readily computed with access to the weight matrix $\mathbf W$. As we do in our experiments, one can also approximate $\mathbf W$ by a low rank matrix via an approximate SVD. We do not prove guarantees when $r > \mathrm{rank}(\mathbf W)$.

---

> > ### Comment · Reviewer_ftz8 · 2023-11-19
> > **Further question on the lower bound for r = O(1)**
> >
> > In the reply, the authors claim that the size of the output also requires $O((n+d)k)$ storage space since it is a factorization. However, since $r = O(1)$, I am not sure whether it is possible that the factorization can be done with even fewer bits of space. For instance, is it possible that the factorization of the $n\times d$ matrix $L$ is sparse? Or is it possible to further factorize $L$ into a $n\times r$ matrix and a $r\times d$ matrix? These seem to be possible to further reduce the bit complexity. Hence, I am asking whether there exists a formal theorem/proof for the bit lower bound.

---

> > > ### Author Response · Authors · 2023-11-19
> > > **Thank you for the question!**
> > >
> > > We thank the reviewer for the question. If $r = 1$, then $\mathbf W$ can be set to the all ones matrix, in which case WLRA becomes the standard low rank approximation problem. A formal information theoretic lower bound for the fact that low rank approximation requires $\Omega((n+d)k)$ bits is standard, and follows from, e.g., applying our lower bound of Theorem 1.5 with $r = 1$. A stronger lower bound against any algorithm which recovers even just an approximate right factor of the SVD is shown in Theorem 82 in Section 10.2 of Boutsidis-Woodruff-Zhong 2016 "Optimal Principal Component Analysis in Distributed and Streaming Models"

---

### Official Review · Reviewer_hd3F · 2023-10-24

**Soundness:** 3 good
**Presentation:** 3 good
**Contribution:** 3 good
**Rating:** 8
**Confidence:** 2

**Summary:**

This paper proposed a new approximate low rank weighted recovery problem, which is very interesting.

**Strengths:**

1. New $\kappa$ approximate framework
2. Both theoretical and practical algorithms are proposed, which are actually simple to use.
3. Theoretical guarantees are offered to ensure the quality of the solution in certain cases.
4. Experiments conducted are convincing.

**Weaknesses:**

1. Writing could be clearer with the different notations, and the overall objective the paper wants to achieve.

**Questions:**

N/A

---

> ### Author Response · Authors · 2023-11-18
> **Thank you for your review!**
>
> We thank the reviewer for their comments.

---

> > ### Comment · Reviewer_hd3F · 2023-11-22
> >
> > Without saying but I'll keep my original score. I think the authors proposed an interesting framework, and its immediate usefulness is not the most important concern in my eye. This is a paper tailored more towards bringing forward a new way of thinking from a theoretical perspective (though incremental in some ways).

---

### Official Review · Reviewer_GrgZ · 2023-10-25

**Soundness:** 3 good
**Presentation:** 3 good
**Contribution:** 2 fair
**Rating:** 5
**Confidence:** 4

**Summary:**

Weighted low rank approximation (WLRA) is an important and fundamental problem in numerical linear algebra, statistics, and machine learning. Specifically, given two matrices $\mathbf{A}$, $\mathbf{W} \in \mathbb{R}_{\ge 0}$, and a parameter $k$, our goal is to minimize $|| \mathbf{W} \circ (\mathbf{A} - \widetilde{\mathbf{A}})||_F$ subject to $\mathrm{rank}(\widetilde{\mathbf{A}}) \le k$. This paper considers one of its relaxed versions: solving a matrix $\widetilde{\mathbf{A}}$ such that $|| \mathbf{W} \circ (\mathbf{A} - \widetilde{\mathbf{A}}) ||_F \le \kappa \cdot \min _{\mathrm{rank}(\mathbf{A}')\le k} ||\mathbf{W} \circ (\mathbf{A} - \mathbf{A}')||$, which assumes $\mathrm{rank}(\mathbf{W}) = r$ and removes the rank bound for matrix $\widetilde{\mathbf{A}}$.
For the above problem, this paper proposes a simple algorithm and proves its correctness. As a corollary, this paper obtains the first relative error guarantee for unsupervised feature selection with a weighted $F$-norm objective. In addition, this paper researches the communication complexity for WLRA and gives the almost matched upper bound and lower bound.

**Strengths:**

(1) This paper proposes a simple algorithm for one relaxed WLRA problem and proves its correctness.
(2) It extends to unsupervised feature selection with a weighted $F$-norm objective.
(3) It explores the communication complexity of the WLRA problem and gives the almost matched upper bound and lower bound.
(4) The experimental results indicate the strengths of the proposed algorithm with respect to the approximation loss and running time, compared with the existing methods.
(5) This paper is well-written and easy to understand.

**Weaknesses:**

(1) This paper relaxes the classical WLRA problem with two conditions: 1) removing the low-rank requirement for matrix $\widetilde{\mathbf{A}}$; 2) assuming the weight matrix $\mathbf{W}$ is low rank. Given these two conditions, the problem becomes much easier, and the proposed algorithm is kind of trivial. Furthermore, if one discarded the low-rank requirement for matrix $\widetilde{\mathbf{A}}$, the relaxed WLRA problem would be kind of insignificant.
(2) The unsupervised feature selection with a weighted $F$-norm objective directly follows Theorem 1.2. Also, although the bounds for communication complexity of WLRA problem are almost tight, the results and processes are kind of straightforward. Therefore, the contributions in this paper are quite limited.
(3) In Related Work, the following reference is missing. $\textit{Recovery guarantee of weighted low-rank approximation via alternating minimization}$.

**Questions:**

See Weaknesses.

---

> ### Author Response · Authors · 2023-11-18
> **Thank you for your review!**
>
> > This paper relaxes the classical WLRA problem with two conditions: 1) removing the low-rank requirement for matrix $\tilde{\mathbf A}$; 2) assuming the weight matrix is low rank. Given these two conditions, the problem becomes much easier, and the proposed algorithm is kind of trivial. Furthermore, if one discarded the low-rank requirement for matrix $\tilde{\mathbf A}$, the relaxed WLRA problem would be kind of insignificant.
>
> Please note that even if we allow $\tilde{\mathbf A}$ to violate the low rank condition, we can still study a non-trivial problem if $\tilde{\mathbf A}$ is structured, i.e. can be stored using much fewer bits of memory or can be applied much quicker than $\mathbf A$. We agree that our proposed algorithm is simple, yet the analysis is nontrivial and our experiments show that our idea leads to great practical benefits over prior work, as demonstrated by our experiments.
>
> > The unsupervised feature selection with a weighted F-norm objective directly follows Theorem 1.2. Also, although the bounds for communication complexity of WLRA problem are almost tight, the results and processes are kind of straightforward. Therefore, the contributions in this paper are quite limited.
>
> The fact that our Theorem 1.2 immediately gives the unsupervised feature selection result shows that Theorem 1.2 is a powerful result. We ask the reviewer to consider the importance of the problem of unsupervised feature selection with a weighted objective, which is a commonly encountered problem in practical data analysis, and the fact that such results were not possible prior to our result.
>
> Our communication complexity lower bound indeed uses standard tools, but we consider the conceptual message resulting from this theorem to be our more important contribution. Our result shows that when considering the communication complexity of the WLRA problem, the rank of the weight matrix must necessarily factor into the communication complexity. That is, without the low-rank assumption on the weight matrix, the WLRA problem is difficult in the communication complexity model. We believe this perhaps surprising message to be one of our most important contributions, as a priori it is not clear the rank would parameterize the communication complexity.
>
> > In Related Work, the following reference is missing: Recovery guarantee of weighted low-rank approximation via alternating minimization.
>
> We have included this work in the Related Work section.

---

> > ### Comment · Reviewer_GrgZ · 2023-11-20
> >
> > Thank you for your feedback. I still have the following concerns.
> > (1) The rank $r$ of the weight matrix $W$ is known in advance and $r = O(1)$. I think this assumption is kind of strong. Given this condition, the space complexity $O( (n+d) rk))$ almost matches the lower bound $\Omega( (n+d) k)$. This cannot be taken as a contribution since in classical WLRA [Li-Liang-Risteski'16, Song-Ye-Yin-Zhang'23], the output matrix has rank $k$ and thus takes space $O( (n+d) k)$.
> > (2) There is no theoretical guarantee for Algorithm 2. You claimed that its effectiveness is demonstrated by experiments, which cannot convince me since it only works for four datasets.
> > (3) This paper does not analyze the time complexity of the proposed algorithm. For an $n \times d$ matrix, it takes time $O(ndrk)$ to compute its rank $rk$ approximation by SVD. However, the existing work [Song-Ye-Yin-Zhang'23] for WLRA has time complexity $O(ndk \log(1/\epsilon))$. Therefore, this paper requires $r = O(1)$, and in terms of space and time, the proposed algorithm has no advantage compared with the existing work on classical WLRA.
> > (4) For column subset selection (Corollary 1.3), the algorithm returns a set $S$ of size $|S| = O(r k / \epsilon)$ and a matrix $X \in \mathbb{R}^{|S| \times d}$, which quantity is taken as a column subset selection of matrix $A$? Is it $(W \circ A)|^S$? If yes, how can $(W \circ A)|^S$ approximate $A$?

---

> > > ### Author Response · Authors · 2023-11-21
> > > **Thank you for the additional questions!**
> > >
> > > (1) We note that the alternating minimization algorithms for WLRA described by Li-Liang-Risteski'16 [LLR16] and Song-Ye-Yin-Zhang'23 [SYYZ23] also have strong assumptions on the input, and also gives a different guarantee, and thus is formally incomporable to our results. These works assume that the ``ground truth'' solution is incoherent, that weight matrix has a spectral gap from the all ones matrix, and that the weight matrix is incoherent. In particular, the algorithm has no guarantees even if $W$ is the all ones matrix, corresponding to the standard unweighted LRA problem. Note that we do not assume any conditions on the input at all, and only impose a rank restriction on the weight matrix. This allows us to handle many important classes of weight matrices not handled by [LLR16, SYYZ23] including a broad class of popular structured matrices (see our response to Question (2)). Furthermore, the recovery guarantee studied by [LLR16] and [SYYZ23] is that two factors $\tilde U$ and $\tilde V$ satisfy the spectral norm guarantee $\lVert A - \tilde U\tilde V\rVert \leq O(k)\lVert W\circ (A - A_k)\rVert + \varepsilon$. This guarantee does not give $(1+\varepsilon)$ relative error guarantees in the original Frobenius norm problem, which is the guarantee that we study in our current work. Thus in summary, our work gives an arguably far stronger $(1+\varepsilon)$ objective function guarantee, with a bicriteria rank $rk$ solution, under the assumption that $\mathbf W$ is rank $r$. On the other hand, [LLR16] and [SYYZ23] give a far larger $O(k)$ factor error guarantee for the spectral norm (which is different from the usual formulation of WLRA), with assumptions of incoherence on the input matrix as well as the weight matrix.
> > >
> > > (2) In Appendix A, we have added a discussion of how a broad family of structured matrices formed by the sum of support-disjoint rank 1 matrices and a sparse matrix can be stored in a way such that the $\mathbf W^{\circ -1}\circ\tilde{\mathbf A}$ of our WLRA algorithm can be applied to a vector and stored efficiently, without forming the matrix $\mathbf W$. This family captures, for example, all of the classes of structured matrices of interest discussed in the prior work of Musco-Musco-Woodruff ``Simple Heuristics Yield Provable Algorithms for Masked Low-Rank Approximation'' including Low-Rank Plus Sparse, Low-Rank Plus Diagonal, Low-Rank Plus Block Diagonal, Monotone Missing Data Pattern, and Low-Rank Plus Banded matrices. Thus, we obtain provable guarantees for many popular weight matrices used in practice, with efficient algorithms. We note that these algorithms are not handled by work of [LLR16, SYYZ23], since, for instance, block diagonal matrices are degenerate.
> > >
> > > We also would like to point out that the other reviewers have viewed our experimental evidence positively ("The experiments suggest that the resulting weighted low-rank approximation is in the range of the state-of-the-art in terms of approximation quality.", "Experiments conducted are convincing.").
> > >
> > > (3) Our algorithm only depends on the primitive of a rank $rk$ SVD, which can be computed up to a $(1+\varepsilon)$ error in the Frobenius objective in $O(\mathsf{nnz}(\mathbf A)) + \tilde O(n r^2k^2\varepsilon^{-4})$ time (Clarkson-Woodruff 2013 "Low Rank Approximation and Regression in Input Sparsity Time"). Thus when $r$ is small enough that $r^2 k \leq d$, then this is faster than the running time of $O(ndk)$ due to [LLR16] and [SYYZ23], for constant $\varepsilon$. In particular, even for sparse matrices with small $\mathsf{nnz}(\mathbf A)$, this algorithm requires $O(ndk)$ time in each iteration, as the alternating minimization algorithm involves solving $d$ instances of linear regression with a dense $n\times k$ matrix. Please also note the difference in guarantees (see the response to Question (1)), so these guarantees are formally incomparable.
> > >
> > > (4) The subset of columns approximates $A$ in the sense that they are a good left factor for a low rank decomposition. That is, if $U$ is the $n\times\lvert S\rvert$ matrix formed by the subset of columns $S$, then $\min_V \lVert W\circ (A - UV)\rVert_F^2 \leq (1+\varepsilon) \mathrm{OPT}$ where OPT is the cost of the optimal rank $k$ solution.

---

> > > > ### Comment · Reviewer_GrgZ · 2023-11-21
> > > >
> > > > Thank you for your clarification.
> > > > (1) Your explanation about the differences with the algorithms based on alternating minimization [LLR'16, SYYZ'23] makes sense.  What is $\kappa$'s value? Is it equal to $1+\epsilon$?
> > > > (2) I don't negate your experimental results. My question is $\textit{could you give the theoretical guarantee of Algorithm 2 but not verification by just running experiments?}$
> > > > (3) Your explanation about the running time makes sense. It would be better to add the time complexity of the proposed algorithm to the future paper.

---

> > > > > ### Author Response · Authors · 2023-11-21
> > > > >
> > > > > (1) Indeed, we can take $\kappa = (1+\varepsilon)$ with running time $O(\mathsf{nnz}(\mathbf A)) + \tilde O(nr^2k^2\varepsilon^{-1})$ by using a randomized low rank approximation algorithms of Avron-Clarkson-Woodruff'17 "Sharper Bounds for Regularized Data Fitting". We have clarified this in Theorem 1.2 in our revised pdf.
> > > > >
> > > > > (2) In Appendix A, we have added provable guarantees for variations on Algorithm 2 for popular structured families of matrices including Low-Rank Plus Sparse, Low-Rank Plus Diagonal, Low-Rank Plus Block Diagonal, Monotone Missing Data Pattern, and Low-Rank Plus Banded matrices. We believe this already gives provable guarantees for a large family of interesting weight matrices. However, in general, we are not able to analyze Algorithm 2 as stated.
> > > > >
> > > > > (3) Thank you for the suggestion! We have added the running time statement in Theorem 1.2 in our revised pdf.

---

> > > > > > ### Comment · Reviewer_GrgZ · 2023-11-21
> > > > > >
> > > > > > Thank you for your feedback.

---

### Official Review · Reviewer_e1o5 · 2023-11-06

**Soundness:** 3 good
**Presentation:** 3 good
**Contribution:** 2 fair
**Rating:** 5
**Confidence:** 4

**Summary:**

In their manuscript, the authors propose an algorithm for the well-known weighted low-rank approximation problem, which is known to be NP-hard, but which has various applications in statistics, signal processing and machine learning. If W is the (non-negative) weight matrix, the authors consider taking the partial SVD of W \circ A, if A is the matrix to be approximated, and then multiply the entrywise inverse of W to the resulting matrix. They provide approximation guarantees for the algorithm as well as a supposedly more practical variant of the algorithm, Algorithm 2, which is meant to avoid a computational pitfall of the original method. Furthermore, they provide an analysis of the resulting communication complexity, and relate that to a lower bound. Finally, they conduct numerical simulations comparing the algorithm's performance on model compression datasets with other state-of-the-art weighted low-rank approximation algorithms. The authors also shed light on the appropriateness of a low-rank assumption on Fisher information matrices in the context of neural network loss functions.

**Strengths:**

The proposed algorithms is conceptually simple and appears to be new. The approximation guarantee of Theorem 1.2 is reasonable and simple. The experiments suggest that the resulting weighted low-rank approximation is in the range of the state-of-the-art in terms of approximation quality.  The presentation of the results is relatively clear and many relevant papers and methods are cited.

**Weaknesses:**

The main motivation of the algorithm as well as the theoretical results are tailored to the case where the weight matrix W is low-rank. However, the fundamental problem in this setting is that there is no reason why the entrywise inverse matrix W^{\circ -1} is low-rank, which leads to the necessity of computing a dense matrix in Algorithm 1, as the authors state, making the algorithm rather inpractical in a large-scale setting. With the practical variant of Algorithm 2, only the storage issue of the resulting approximation is mitigated, but not not the fact that W^{\circ -1} needs to be computed in a dense manner, which requires O(nd) of storage in intermediate calculations.
While the communication complexity discussion of Section 1.1.2 is interesting, it is unclear whether and how the algorithm can be implemented efficiently in a distributed manner. While the authors claim that their result is the first about communication complexity for weighted low-rank approximation problems, it seems that communication complexity for such results was already previously discussed, e.g., in Musco, Cameron, Christopher Musco, and David Woodruff. "Simple Heuristics Yield Provable Algorithms for Masked Low-Rank Approximation." _Innovations in Theoretical Computer Science Conference (ITCS 2021)_. 2021.

**Questions:**

- Can you discuss the time complexity of the resulting algorithms and intermediate space complexity of Algorithm 1 and Algorithm 2?
- Please discuss your result in the context of previous communication complexity results.
- I did not find the code for the timing experiments in the provided code submission. Furthermore, it would be good if the hyper parameter choices of the reference algorithms are provided in the manuscript, e.g. of adam and em.

---

> ### Author Response · Authors · 2023-11-18
> **Thank you for your review!**
>
> > With the practical variant of Algorithm 2, only the storage issue of the resulting approximation is mitigated, but not not the fact that $\mathbf W^{\circ -1}$ needs to be computed in a dense manner, which requires $O(nd)$ of storage in intermediate calculations.
>
> In Appendix A, we have added a discussion of how a broad family of structured matrices formed by the sum of support-disjoint rank 1 matrices and a sparse matrix can be stored in a way such that $\mathbf W^{\circ -1}$ can be applied and stored efficiently, without forming the matrix $\mathbf W$. This family captures, for example, all of the classes of structured matrices of interest discussed in the prior work of Musco-Musco-Woodruff "Simple Heuristics Yield Provable Algorithms for Masked Low-Rank Approximation" including Low-Rank Plus Sparse, Low-Rank Plus Diagonal, Low-Rank Plus Block Diagonal, Monotone Missing Data Pattern, and Low-Rank Plus Banded matrices.
>
> > While the communication complexity discussion of Section 1.1.2 is interesting, it is unclear whether and how the algorithm can be implemented efficiently in a distributed manner.
>
> Our algorithm essentially only relies on the primitive of an SVD, which is known to be efficiently implementable in a distributed setting (see, e.g., Boutsidis-Woodruff-Zhong 2016 "Optimal principal component analysis in distributed and streaming models"). Our work shows that this primitive is indeed useful for weighted low rank approximation, especially for structured weight matrices (see Response 1)x, and that this algorithm is nearly optimal for certain distributed formulations of the weighted low rank approximation problem.
>
> > Can you discuss the time complexity of the resulting algorithms and intermediate space complexity of Algorithm 1 and Algorithm 2?
>
> Both Algorithms 1 and 2 involve computing rank $rk$ approximations to matrices of size $n\times d$. Exact SVD can be computed in $O(nd^{\omega-1})$ operations using $O(nd)$ words of space, whereas randomized approximation algorithms can approximate the Frobenius objective up to a $(1+\varepsilon)$ factor in time and space $O(\mathsf{nnz}(\mathbf A)) + \tilde O(n) \mathrm{poly}(k/\varepsilon)$, where $\mathsf{nnz}(\mathbf A)$ denotes the number of nonzero entries of $\mathbf A$ (Clarkson-Woodruff 2013). Thus for dense matrices where $\mathsf{nnz}(\mathbf A) = O(nd)$, the intermediate calculations do not affect the space complexity of the algorithm. We note that in the setting of machine learning applications, dense linear algebra is often used due to improved hardware utilization, and thus $\mathsf{nnz}(\mathbf A) = nd$ is a common setting.
>
> > Please discuss your result in the context of previous communication complexity results.
>
> The study of communication complexity of WLRA is, to the best of our knowledge, new in this work. The work by Musco-Musco-Woodruff mentioned by the reviewer indeed discusses communication complexity and WLRA together. However, in that work, communication complexity is used as a tool to analyze the "complexity" of the weight matrix $\mathbf W$, and the analysis results in an offline algorithm for WLRA. That is analogous to our use of the rank of $\mathbf W$ as a tool to analyze the ``complexity'' of $\mathbf W$. On the other hand, that work does not address the problem of how a solution to the WLRA problem can be communicated between servers, which is the topic of our work.
>
> > I did not find the code for the timing experiments in the provided code submission. Furthermore, it would be good if the hyper parameter choices of the reference algorithms are provided in the manuscript, e.g. of adam and em.
>
> The code for the timing experiments was provided in `weighted_lra.py` in the provided code submission. The adam algorithm was run with 100 epochs, with an initial learning rate of 1.0 that is decayed by a factor of 0.7 for every 10 steps. The em algorithm was run for 25 epochs. These details have been included in the manuscript.

---

> ### Comment · Reviewer_e1o5 · 2023-11-23
>
> >> With the practical variant of Algorithm 2, only the storage issue of the resulting approximation is mitigated, but not not the fact that $\mathbf W^{\circ -1}$ needs to be computed in a dense manner, which requires $O(nd)$ of storage in intermediate calculations.
>
> >In Appendix A, we have added a discussion of how a broad family of structured matrices formed by the sum of support-disjoint rank 1 matrices and a sparse matrix can be stored in a way such that $\mathbf W^{\circ -1} $ can be applied and stored efficiently, without forming the matrix $\mathbf{W}$. This family captures, for example, all of the classes of structured matrices of interest discussed in the prior work of Musco-Musco-Woodruff "Simple Heuristics Yield Provable Algorithms for Masked Low-Rank Approximation" including Low-Rank Plus Sparse, Low-Rank Plus Diagonal, Low-Rank Plus Block Diagonal, Monotone Missing Data Pattern, and Low-Rank Plus Banded matrices.
>
> The special cases you are pointing out are problematic. First, I'd like to clarify for the other reviewers that unlike your comment suggests, based on my reading of your updated appendix, the suggested procedure does not apply to all matrices within the families of Low-Rank Plus Sparse matrices etc., but only to specific cases of such matrices. Furthermore, these special cases are all about matrices that have a significant number of zero entries. What does $\mathbf{W}^{\circ -1}$ even mean in these cases, as one is essentially dividing by zero?
>
> Furthermore, I'd like to make my original concern more precise: Algorithm 2 will still require the computation of dense matrices as the second step of Algorithm 2 requires the multiplication of (tilde) $\mathbf W_{inv}^{\circ -1} $ with the entries of $\mathbf{A}$, where (tilde)$\mathbf W_{inv}$ is the rank-$rk$ approximation of $\mathbf{W}^{\circ -1}$. Even if $\mathbf{W}^{\circ -1}$ were somehow to be computed efficiently, there is no reason why the entrywise inverse (tilde)$\mathbf W_{inv}^{\circ -1}$ of (tilde)$\mathbf W_{inv}$ can be computed without dense matrix processing. My understanding is that (tilde)$\mathbf W_{inv}$ will not inherit structural properties of $\mathbf W$ or of $\mathbf W^{\circ -1}$.
>
> Thank you for addressing my other questions. In any published version of this paper, it would be important to make the distinction between communication complexity notions of the your paper and the Musco-Musco-Woodruff paper.
>
> Also in the view of the discussion of other reviewers' comments, I maintain my rating for the submission.

---

> ### Author Response · Authors · 2023-11-23
>
> We believe there are several inaccuracies in the reviewer's response, and hope the reviewer can consider their evaluation in light of our responses below:
>
> (1) It is inaccurate to say that all of the practical weight matrices we handle have a large number of zero entries. A simple (and uninteresting) special case is a rank-1 matrix, which could have an arbitrary density of zeros or ones, or even be all 1s, and so  has no zero entries in that case. Also low-rank-plus diagonal or low rank plus banded is mostly ones, not zeros. Also (and more interestingly) low rank plus block diagonal as well as monotone missing data patterns can have an arbitrary density of zeros and ones depending on the block or prefix sizes.
>
> (2) Note that if $\mathbf W_{i,j} = 0$, then we can set $\mathbf W^{\circ -1}_{i,j}$ to be arbitrary (say, always equal to $1$ for concreteness) since such entries will be zero-ed out when we apply $\mathbf W$ entrywise, so in particular there is no worry of dividing by zero if that is the concern. This can be seen in the first line of the proof of Theorem 1.2 at the top of page 6.
>
> (3) We cannot prove that Algorithm 2 works for every low rank plus sparse matrix - however, we believe we make a significant theoretical advance by applying it to a large class of popular weight matrices used in practice and giving the first theoretical bounds for such matrices. In Appendix A, for each weight matrix described there, either $\mathbf W^{\circ -1}$ already has low rank, or one can apply $\mathbf W^{\circ -1}$  quickly as we describe there. Note that if $\mathbf W = u \cdot v^T$, then $\mathbf W^{\circ -1} = u^{\circ -1} \cdot (v^T)^{\circ -1}$, where zero entries of $u$ or $v$ can be replaced with an arbitrary number (e.g., $1$ for concreteness) for the reason described in (2) above. Thus, $\mathbf W^{\circ -1}$ does partially inherit low rank structure from $\mathbf W$. Also, as noted in Appendix A, this inheritance of low rank structure holds for larger-rank matrices $\mathbf W$ provided the combinatorial rectangles defined by the supports of the low rank factors are disjoint, which together with our ability to handle sparsity, solves five different previously studied and practical weight patterns - in fact for every motivating weight matrix studied in Musco-Musco-Woodruff we give an efficient algorithm. Besides our theoretical results, which at least to us are surprising since a priori $\mathbf W^{\circ -1}$ may not look easy to apply, we give experimental results for an even wider class of weight matrices $\mathbf W$ by heuristically computing a low rank approximation of $\mathbf W^{\circ -1}$.
>
> (4) We will stress that in the Musco-Musco-Woodruff paper, there is no communication or distributed setting at all. The notion of ``communication complexity" is a combinatorial notion about a matrix, and that is used to describe the rank of their bicriteria approximations. In our submission, however, we describe a distributed setting and use communication complexity to mean the maximum length of a protocol transcript, over all inputs and random choices.
>
> We are happy to make these points clearer in the writing in case there is any confusion. Please let us know if there are any other questions, and thanks for giving us the opportunity to respond.

---

### Meta-Review · Area_Chair_rP3Y · 2023-12-06

**Metareview:**

The paper considers the problem of finding a low rank matrix that approximates a given matrix where the errors in different entries are weighted differently. The paper focuses on the special case where the weight matrix has low rank and introduces a new definition for the approximate solution: the approximate solution does not need to have low rank but only needs to have a small description length. The paper gives an algorithm for this task with experiments showing the good performance compared with previous methods for this setting.

On the downside, several reviewers question the new definition, noting that it makes the problem "kind of trivial" and inconsistent with the original objective. However, one reviewer strongly argues for the paper and note that the definition is a novel viewpoint. One reviewer also notes that the setting where the new algorithm applies seem narrower than just low rank weight matrix because it also requires that computing the low rank approximation of the entrywise inverse of the weight matrix can be done efficiently. It is less clear what this condition entails.

**Justification For Why Not Higher Score:**

3 out of 4 reviewers recommend reject and the sole supporter of the paper unfortunately has a very terse review. It is not clear when the new algorithm applies because the condition seems to be more than just a low rank weight matrix. Furthermore, several reviewers are concerned that the new definition is inconsistent with the original objective.

**Justification For Why Not Lower Score:**

N/A

---

### Decision · Program_Chairs · 2024-01-16

Reject